# Changes in Quality Characteristics and Metabolite Composition of Low-Temperature and Nitrogen-Modified Atmosphere in Indica Rice during Storage

**DOI:** 10.3390/foods13182968

**Published:** 2024-09-19

**Authors:** Yanan Zhao, Yanfei Li, Zhigang Gong, Xuguang Liu, Haoxin Lv, Yan Zhao

**Affiliations:** 1School of Food and Strategic Reserves, Henan University of Technology, Zhengzhou 450001, China; 2022920080@stu.haut.edu.cn (Y.Z.); liyanfei@haut.edu.cn (Y.L.); lvhaoxin0129@126.com (H.L.); 2SinoGrain Tongling Direct Depot Co., Ltd., Tongling 244000, China; gongzhigang2024@126.com (Z.G.); liuxuguang2024@yeah.net (X.L.)

**Keywords:** nitrogen-modified atmosphere, low temperatures, quality, rice, non-targeted metabolomics

## Abstract

A low temperature (LT) is used to delay grain deterioration effectively. In practical applications, a nitrogen-modified atmosphere (N_2_) is also an effective way of preventing grain pests and delaying grain deterioration. However, there are few studies on grain quality changes using a combination treatment of an LT and N_2_ during storage. In this study, the storage quality, processing characteristics, and metabolites of rice under conventional storage (CS), LT (20 °C), N_2_ (95%), and LT+N_2_ treatments were analyzed for 180 days, under a controlled humidity of 65% ± 2%. The results showed that compared to the CS, LT, and N_2_ treatments, the LT+N_2_ treatment had the best effect in retarding the increase in MDA and electrical conductivity and deferring the decrease in CAT activity. In addition, the LT+N_2_ treatment maintained the color of the rice better and sustained a better processing quality. Non-targeted metabolomics analysis further confirmed that the LT+N_2_ treatment maintained the vigor of the rice and retarded its spoilage by activating the metabolisms of amino acids, carbohydrates, and flavonoids. These results suggest a favorable practice for preventing storage deterioration and increasing the processing quality for rice storage. They provided new insights into the mechanisms of rice quality changes using the combination treatment of an LT and N_2_.

## 1. Introduction

Rice (*Oryza sativa* L.) is one of the main cereals grown in the world [1]. Rice is the primary source of energy for humans. The FAO’s forecast for world cereal production in 2023 is that global rice production is expected to reach 526.2 million tons (milled basis) in 2024, up 0.4 percent from the revised 2023 level. As a result, the task of storing grain crops increases dramatically [2]. However, during storage, rice is vulnerable to heat, mold, and sprouting, leading to poor quality and loss of vitality, resulting in significant losses. The quality of stored rice deteriorates principally due to poor storage conditions [3]. Therefore, it is essential to seek the appropriate storage conditions to delay rice quality deterioration during grain storage.

The storage quality of rice is influenced by the temperature, moisture content, and the composition of gases between grain kernels in general. Among them, the storage temperature could influence the enzymatic activities and protein, starch, and lipid contents of stored rice [4]. For example, it was indicated that high temperatures increased the content of fatty acids and lipid oxidation [3]. The increase in fatty acid values in rice stored at high temperatures may be due to the increase in free fatty acids, which resulted in the increased deterioration of the paddy. A decrease in soluble sugars, which may be related to endogenous amylase activity, occurs with prolonged storage at high temperatures. A low temperature (LT) hampers protein hydrolysis and amino acid solubilization, resulting in reduced protein and starch digestibility [5]. As the temperature increases, the proteins transform from an ordered to a disordered structure. This increased disorder exposes hydrophobic residues on the surface of protein molecules, which may result in poorer starch gelatinization and reduced eating quality [6]. Thus, LT storage is more beneficial for maintaining rice quality, inhibiting microbial growth, and delaying spoilage.

The disinfestation of insects in grains is an essential process during the storage of grains. Modifying the atmosphere is a widely used practice for green grain storage that can effectively slow down the respiration of grain seeds and pests. The disinfestation process of a modified atmosphere (MA) involves changing the environmental conditions of stored grains by adjusting the concentrations of gases (O_2_, N_2_, and CO_2_), relative humidity (RH), and atmospheric pressure [7]. Storage microorganisms and insects require oxygen to support their life activities, and low oxygen concentrations cause the life activities to be suppressed and delay the deterioration of the physicochemical properties of the grains [8]. Therefore, the trend in current research is toward the use of a MA, either through CO_2_ enhancement or O_2_ reduction [9]. CO_2_-MA storage was found to be effective at enhancing the antioxidant capacity of rice, preserving enzyme activities, and reducing the loss of characteristic aroma compounds at a concentration of 35% CO_2_ [10]. A N_2_-MA could effectively delay rice quality deterioration and improve the oxidative stress capacity of rice [11]. Compared with a CO_2_-MA, a N_2_-MA is applied more widely in a granary because of its low cost, easy preparation, and security [12]. Liu et al. (2022) [13] demonstrated that storage pests were effectively controlled at a 95% nitrogen concentration. Therefore, a N_2_-MA with a nitrogen concentration of 95% is an up-and-coming alternative to the traditional use of chemicals for the control of storage pests. Although the N_2_-MA under an LT appeared to be more effective at preserving the physical, chemical, and technological characteristics during bean storage [14], there have been fewer studies on the LT+N_2_ treatment being applied to rice.

Non-targeted metabolomics (NTM) is the hypothesis-generating, global, and unbiased analysis of all small molecule metabolites present in a biological system under given conditions [15]. NTM approaches can detect a wide range of metabolites and provide comprehensive insights into metabolite profiles of biological samples. As a result, they have been used to carry out research in medicine, botany, and microbiology [16]. In previous studies, NTM was used to investigate the mechanism of yellowing in rice grains [17]. However, there was no study on the differences in metabolites during rice storage using NTM, especially for the combination of LT and N_2_ treatments. Therefore, it is essential to investigate the effects of the LT+N_2_ treatment on the deterioration of rice quality, its impact on storage and the processing quality, as well as the metabolic mechanisms involved. In this study, an NTM approach using UHPLC-OE-MS was used to investigate the differentially expressed primary metabolites of terminal rice kernels under LT, N_2_-MA, and LT+N_2_-MA treatments and to compare the differences in the primary metabolites.

This study aimed to elucidate the effect of the storage temperature and gas environment on rice quality deterioration using the NTM approach. Rice was stored under different conditions for 180 days, and these parameters, including the fatty acid value (FAV), malondialdehyde content (MDA), electrical conductivity, catalase activity (CAT), pasting properties, color, and NTM, were investigated in “Huanghuazhan” rice following the CS, LT, N_2_, and LT+N_2_ treatments. We utilized NTM to analyze rice stored for 180 days under different conditions, and metabolic pathway analysis was performed based on the differential metabolites obtained. The purpose of this study was to provide a scientific theoretical basis for the storage of rice in different storage environments.

## 2. Materials and Methods

### 2.1. Rice Materials

In November 2022, rice of the cultivar “Huanghuazhan” was harvested in Wuhan, China. By following the method of Bilhalva et al. [18], the moisture content of the rice was calibrated with an accuracy of 14.5 ± 0.5%. 

### 2.2. Rice Storage 

In this experiment, four storage conditions were used: CS, LT, N_2_, and N_2_+LT. CS: The rice samples were stored in 0.24 mm polyethylene bags at 30 ± 1 °C. LT: The rice samples were kept in 0.24 mm polyethylene bags at 20 ± 1 °C. N_2_: The rice samples were stored in 0.24 mm thick polyethylene bags (95% nitrogen) at 30 ± 1 °C. LT+N_2_: The rice samples were stored in 0.24 mm polyethylene bags (95% nitrogen) at 20 ± 1 °C. The temperature conditions and a maintained humidity of 65% ± 2% were controlled using an incubator. The residual oxygen concentration was the result of a gas detector measurement (MIC-600, Shenzhen Yiyuntian Technology Co., Ltd., Shenzhen, China; detection range 0~100%; accuracy 0.1%). Residual oxygen levels in the packaging were measured weekly and maintained at less than 5.0 ± 0.1% during storage. The rice was stored for 180 days, and samples were taken every 30 days to determine the rice quality.

### 2.3. Fatty Acid Value (FAV)

The FAV was measured according to Zhan’s method [19], with a few minor modifications. The rice samples were dehulled using a dehuller (BLH-3250B, Bethlehem Instruments Ltd., London, UK). The husked rice samples were pulverized using the following mill: TDW-5000, Beijing Tongxin Tianbo Technology Development Co., Ltd., Beijing, China. Anhydrous ethanol was used for mixing with rice flour, which was then shaken and filtered. Carbon dioxide-free distilled water was mixed with the filtrate (the filtrate in the blank experiment was replaced with anhydrous ethanol), and titration was performed after adding 4 drops of phenolphthalein as an indicator. The filtrate was diluted and titrated with standard potassium hydroxide. The FAV was calculated as the volume of potassium hydroxide required to neutralize the free fatty acids in 100 g of rice sample. The data were expressed graphically as mg/100 g.

### 2.4. Malondialdehyde (MDA) Content 

The MDA content was determined following the method of Li et al. [20], with a minor modification. Here, 1.00 g of dehulled rice samples were ground and homogenized with 5 mL of 10% trichloroacetic acid (TCA). The mixture was centrifuged at 10,000× *g* for 15 min at 4 °C. Then, the supernatant was collected and mixed with 0.67% trichloroacetic acid (TBA). After heating for 15 min in a boiling water bath, the mixture cooled rapidly. The MDA content was calculated by determining the absorbance rates at wavelengths of 450 nm, 532 nm, and 600 nm. The result was expressed as mg/100 kg.

### 2.5. Electrical Conductivity

The electrical conductivity test was performed using a conductivity meter (DDS307A, Shanghai Yidian Scientific Instrument Co., Ltd., Shanghai, China) following Granella’s method [21], with a small number of modifications. Seeds were weighed at random, unbroken, and filled, of similar size. Rice seeds and deionized water were mixed in a flask and placed in a thermostat at 30 °C for 13 h. Electrical conductivity was measured with a conductivity meter, and the values were expressed in terms of μS/cm. 

### 2.6. Catalase (CAT) Activity

The CAT activity was measured using the CAT kit (SKU: BC0200, Beijing Solarbio Science & Technology Co., Ltd., Beijing, China). The activity of CAT was calculated using the rate of hydrogen peroxide decomposition. The CAT activity was expressed by determining the decrease of 0.01 per minute at 240 nm in the test solution. The data were analyzed in terms of mg/g.

### 2.7. Rice Color Measurement

The rice color was measured with a CR-410 colorimeter (KONICA MINOLTA, JAPAN). The colorimeter was first calibrated with a whiteboard, and then the rice was measured to obtain the values of L*, a*, and b*. The results of the rice color measurement were reported as L*/(lightness), a*/(red-green), and b*/(yellow-blue) values [22]. Chromatic aberration ΔE* was calculated using the following equation:$$\Delta E*=\sqrt{{\left(∆L*\right)}^{2}+{\left(∆a*\right)}^{2}+{\left(∆b*\right)}^{2}}$$

### 2.8. Pasting Properties of Rice

The rice was milled and then passed through a 40-mesh sieve. The pasting properties of rice flours were established using the Rapid Visco Analyzer (RVA) (RVA-TecMaster, Perten Instruments, Inc., Stockholm, Sweden), and a mass of 12% double-distilled water, calculated using RVA software, was added to 3.0 g of rice flour in an aluminum RVA test tube. The rice flour method’s program was used with a total process time of 12.5 min. In short, the temperature–time conditions were held at 50 °C for 1 min; after that, the heating step increased from 50 to 95 °C at 12 °C/min, was held at 95 °C for 2.5 min, then cooled to 50 °C for 3.8 min at 12 °C/min, and held at 50 °C for 1.4 min. The stirrer was maintained at an initial speed of 960 rpm/min for 60 s, then reduced to 160 rpm/min until the test ended. The parameters that were considered include the peak viscosity, peak time, final viscosity, hold viscosity, breakdown viscosity, and setback of the rice samples and were recorded and calculated using the RVA curves reported in cP or °C units.

### 2.9. UHPLC-OE-MS Conditions for Metabolomics Analysis

#### 2.9.1. Extraction of Metabolites

The samples were immediately stored in a −80 °C refrigerator after collection until assay. A total of 20 mg of sample was transferred to the homogenization beads, and 1000 μL of extraction solution (methanol/acetonitrile/water = 2:2:1 (*v*/*v*/*v*)) was added. The extract was vortexed with the isotopically labeled internal standard and sonicated in an ice-water bath for 5 min, and the process 3 times was repeated. After stewing at −40 °C for 1 h, the mixture was centrifuged at 12,000 rpm under 4 °C for 15 min. The supernatant was collected and injected into the plastic autosampler vial. All samples were mixed with an equal amount of supernatant to form a QC sample for testing. 

#### 2.9.2. Non-Targeted Metabolomics Analysis

UHPLC-OE-MS (Vanquish, Thermo Fisher Scientific, Waltham, MA, USA) was used for the non-targeted metabolomics analysis of rice grain samples at the end of storage. The target compounds were separated using liquid chromatography on a Phenomenex Kinetex C18 column (2.1 mm × 50 mm, 2.6 μm) coupled to an Orbitrap Exploris 120 mass spectrometer (Orbitrap MS, Thermo). The mobile phase consisted of 0.01% acetic acid in water as mobile phase A and a mixture of IPA and ACN (1:1, *v*/*v*) as mobile phase B. The autosampler temperature was 4 °C, and the injection volume was 2 μL. The Orbitrap Exploris 120 mass spectrometer was used for its capability to acquire MS/MS spectra in information-dependent acquisition (IDA) mode under the control of the acquisition software (Xcalibur, Thermo). In this mode, the acquisition software continuously evaluates the full-scan MS spectrum. The ESI source conditions were set as follows: sheath gas flow rate as 50 Arb, aux gas flow rate as 15 Arb, capillary temperature of 320 °C, full MS resolution as 60,000, MS/MS resolution as 15,000, collision energy (SNCE) as 20/30/40, and spray voltage as 3.8 kV (positive) or −3.4 kV (negative), respectively [23,24].

#### 2.9.3. Data Preprocessing and Annotation

The raw data were converted to the mzXML format using ProteoWizard and processed with an in-house program, which was developed using R and based on XCMS, for peak detection, extraction, alignment, and integration. The R package and BiotreeDB (V3.0) were applied for metabolite identification [25].

### 2.10. Statistical Analysis

Each experiment was carried out three times in parallel. Statistical analysis was performed using one-way ANOVA and Duncan’s multiple extreme difference test with SPSS (version 20.0, SPSS Inc., USA) to analyze rice sample quality index data. *p* < 0.05 was considered statistically significant. Origin 10.1 (Origin Lab Corp., Northampton, MA, USA) was used for data analysis. The data were expressed as the mean ± standard deviation (SD).

The experiments on the metabolism of rice using UHPLC-OE-MS were carried out six times in parallel. Principal Component Analysis (PCA), Orthogonal Partial Least Squares Discriminant Analysis (OPLS-DA), Hierarchical Cluster Analysis (HCA), and metabolic pathway analysis were performed on the raw data of metabolites using Metaboanalyst 6.0 (https://www.metaboanalyst.ca/, accessed on 1 April 2024). The metabolites in rice varied depending on the storage conditions. The different metabolites were identified and matched to their corresponding metabolic pathways using the Oryza sativa japonica database in the Kyoto Encyclopedia of Genes and Genomes (KEGG).

## 3. Results and Discussion

### 3.1. The Quality Changes in Rice during Storage

The FAV is a measure of the free fatty acid content of rice grains. The changes in the FAV of rice grains under different storage conditions are shown in Figure 1A. The FAV levels increased with time, probably because of the oxidative rancidity of the lipids in the rice kernels during storage. The quality of the rice grain was reduced due to the rancidity of the lipids. Compared to the CS, both the N_2_ and LT treatments caused a significant decrease in the FAV of rice during storage. The LT+N_2_ treatment prevented the increase in free fatty acids and further oxidation of rice during storage and better retarded the increase in the FAV than the N_2_ and LT alone, thus delaying the deterioration of rice quality. 

The lipids found in rice were hydrolyzed, resulting in the production of free fatty acids. The fats were hydrolyzed by lipolytic enzymes to produce peroxides. The peroxides were broken down into smaller molecules, resulting in the formation of aldehydes and ketones, the main product being MDA. Excessive free radicals and MDA accumulate in the mitochondria and cause severe damage to the structure and function of the mitochondrial membrane, exacerbating seed degradation [26]. Figure 1B depicted that the overall MDA of rice showed a decreasing trend, and either N_2_ or an LT could significantly reduce the MDA content of rice. The influence of the LT on rice MDA was more pronounced than that of N_2_, while rice degradation was more effectively inhibited by the combination treatment of LT+N_2_.

As shown in Figure 1C, both an LT and N_2_ retarded the increase in electrical conductivity and deterioration of rice. The integrity of the cell membrane was mirrored by the electrical conductivity. Electrical conductivity was used as an indicator of rice degradation and decreased cell viability. In Figure 1C, the treatment of LT+N_2_ was able to better preserve the integrity of rice cells and maintain seed viability.

CAT is considered a vital ROS-eliminating enzyme that catalyzes H_2_O_2_ into H_2_O and O_2_, preventing the excessive accumulation of H_2_O_2_, which damages the membrane. In our study, as observed in Figure 1D, with the storage time increasing, the CAT activity of rice decreased in the four groups. Both an LT and N_2_ delayed the reduction in the CAT activity of rice, while the combination treatment of an LT and N_2_ was found to better preserve the CAT activity of rice compared to the LT and N_2_ treatments alone.

As shown in Figure 1E–H, the skin color of rice gradually turned dark and red with the extension of the storage time. The L* (brightness) of all the rice samples in the four groups slightly decreased with an extended storage time. In the end, at a storage time of 90 days, the L* values of the LT+N_2_, LT, and N_2_ treatments were all higher than the CS in the rice; the LT alone and the combination treatment showed better high-level brightness than the CS and N_2_ treatments. The total color difference is represented by ΔE∗. The rice stored following LT+N_2_ treatment showed significantly lower values of a* and ΔE∗ from 60 to 180 days of storage, indicating that the combination treatment was more effective at slowing down the darkening of rice during storage.

The effects of the storage temperature and nitrogen gas conditioning on the pasting properties of rice during storage are plotted in Figure 2A–F. The measurements of the rice samples increased with time during storage, including the final viscosity, holding viscosity, peak time, and setback. The peak viscosity and final viscosity followed a rising and then falling trend but with different times for the highest value. The peak time, final viscosity, and peak viscosity were impaired significantly by N_2_. There was no significant effect from the N_2_ setback, holding viscosity, and breakdown viscosity. However, low-temperature treatments (LT and LT+N_2_) significantly reduced the setback, holding viscosity, and breakdown viscosity of rice grains. The peak times changed significantly at 150 days for N_2_ action. Overall, the LT+N_2_ treatment was found to have a better effect on the pasting properties of rice.

In total, the combined effect of LT and N_2_ was more effective in the maintenance of rice grain quality after 180 days of storage.

### 3.2. Metabolite Analysis of Rice under Different Storage Conditions

To validate the effects of an LT, N_2_, and the combined effect of an LT and N_2_ on the metabolites of rice grains, UHPLC-OE-MS was conducted to identify the metabolites in the rice samples. The metabolites that were identified included amino acids, carbohydrates, fatty acids, and flavonoids.

#### 3.2.1. Principal Component Analysis (PCA)

PCA converts the original data into a few linearly independent principal components through orthogonal transformation and dimensionality reduction processing. PCA simplifies the data structure and improves the data processing efficiency. PC 1 and PC 2 together explained 81.6% of the sample variance, PC 1 explained 63.1% of the variance, and the four groups were separated in PC 1. All samples were within the 95% confidence interval, indicating that the PCA results were credible and reproducible. The PCA results indicated that an LT and N2 affected the metabolites of rice.

#### 3.2.2. Orthogonal Projections to Latent Structures Discriminant Analysis (OPLS-DA)

OPLS-DA involves a statistical method of supervised discriminant analysis. It screens for differential metabolites, predicts sample categories, visualizes group separation, and identifies significantly altered metabolites. Each scatter plot in the figure represents a sample. Figure 3C shows that PC1 explains 29.6% of the sample variance, while in Figure 3D, PC 1 explains 29.9% of the sample variance, and in Figure 3E, PC 1 explains 21.5% of the sample variance. These results suggest that different storage conditions have a significant effect on the rice samples. The OPLS-DA analysis was consistent with the PCA analysis. The quality of the model was assessed using the response ordering test. The interpretation ability R2Y (cum) and predictive power Q2 (cum) of the fitted test samples were both close to 1, indicating high predictability in the result.

#### 3.2.3. Identification of Key Metabolites in Rice

The variable importance in the projection (VIP) of the first principal component in the OPLS-DA analysis was obtained to summarize the contribution of each variable to the model. The differential metabolites were screened according to VIP > 1, *p* < 0.05, and FC > 2 and are presented as volcano plots. In CS vs. N_2_, 18 metabolites were up-regulated, and 2 metabolites were down-regulated. In CS vs. LT, 70 metabolites were up-regulated, and 63 metabolites were down-regulated. Finally, in CS vs. LT+N_2_, 72 metabolites were up-regulated, and 75 metabolites were down-regulated (shown in Figure 4).

The heatmap displays the differences in metabolites within groups more effectively. Appendix A illustrate the 50 differential metabolites with the highest *p*-values between groups. The results indicate the differential metabolites for this group comparison based on horizontal and vertical coordinates. 

As illustrated in Appendix A, these results demonstrate that the addition of a low-temperature treatment (LT and LT+N_2_) was more effective than the N_2_ treatment in terms of improving the quality of rice. The quantity and nature of differential metabolites exhibited considerable variability between the treatment groups.

#### 3.2.4. Overview of Differential Metabolites

Through additional comparative analysis of the CS vs. LT+N_2_, CS vs. LT, and CS vs. N_2_ data, Figure 5A shows that there are 140 important metabolites with overlap (Appendix A). Of these, 60 metabolites had the same trend of change, 56 metabolites were down-regulated, and 4 metabolites were up-regulated. In the comparison of CS vs. LT+N_2_ and CS vs. LT, 109 metabolites exhibited the same trend, with 105 metabolites being down-regulated and 4 being up-regulated. Furthermore, 71 metabolites showed the same trend in the comparison of CS vs. LT+N_2_ and CS vs. N_2_, with 62 metabolites being down-regulated and 9 being up-regulated. The study suggests that storage conditions can affect the metabolites of rice grains.

In our experiment, the principal differential metabolite species were organooxygen compounds and saccharides, fatty acids and conjugates, small peptides, flavonoids, and phenolic acids. In contrast to the CS group, the majority of organooxygen compounds, such as saccharides and small peptide substances, were found to be up-regulated by a low temperature and N_2_-MA. Conversely, the majority of fatty acid and conjugate substances were down-regulated. The elevation of flavonoids was effectively inhibited by a low temperature and N_2_-MA. The low-temperature treatment was found to be effective at inhibiting the elevation of phenolic acids, in contrast to N_2_-MA, which was observed to have no significant effect on phenolics.

#### 3.2.5. Overview of Pathway Analysis

The impact of the pathway in which the alterations resulting from the experimental conditions are located was analyzed using Pastway Analysis in Metabo Analyst. The compounds screened were analyzed in detail by projecting them into the KEGG pathway database; the results of the metabolic pathway analysis are presented in a bubble plot, where each bubble represents an identified metabolic pathway (shown in Figure 5). In order to investigate the effects of different conditions on metabolic pathways in rice, the main study was conducted on the following: galactose metabolism; pentose phosphate pathway; alanine, aspartate, and glutamate metabolisms; flavone and flavonol biosynthesis; cyanoamino acid metabolism; and D-amino acid metabolism. These pathways are mainly linked to the metabolisms of amino acids, carbohydrates, and antioxidant compounds.

### 3.3. Analysis of Metabolic Pathways

#### 3.3.1. Amino Acid Metabolism

Amino acids are essential for the adaptation of plants to environmental stress, being required for many specialized metabolic processes. Amino acids are the primary metabolites that directly and indirectly contribute to ROS scavenging [27]. The analysis of the pathway shows that the modified amino acids were mainly associated with the metabolic pathways of alanine, aspartate and glutamate; cyanoamino acid; and D-amino acid (Figure 6). 

Amygdalin is soluble in water. Water can regulate amygdalin synthesis during almond seed germination at the late stage by activating amygdalin synthesis genes and reducing the inhibitory effect of amygdalin on germination [28]. The germination rate of rice decreases as the storage temperature increases, leading to reduced lipid oxidation and changes in the pasting properties of the rice. Additionally, seed viability was negatively impacted. The germination rate of rice decreases as the storage temperature increases, leading to reduced lipid oxidation and changes in the pasting properties of the rice [29]. Under low-temperature conditions, the level of amygdalin significantly increased, which contributed to enhancing the germination rate of rice, thereby maintaining seed vitality and longevity. In comparison, N_2_ treatment alone did not significantly alter the level of amygdalin.

Some amino acids have antioxidant properties. These may synergistically enhance the activities of phenolic compounds. For example, L-asparagine and glutamate scavenge free radicals, reduce hydroperoxides, chelate pro-oxidant transition metals, and even act as biomarkers of oxidative stress [30]. Rice seeds undergo metabolic changes in response to temperature fluctuations. In our study, the LT treatment induced higher levels of L-asparagine and glutamate to enhance the antioxidant capacity in rice during storage. GABA was mainly derived from L-glutamine. It was produced by the catalysis of glutamic acid decarboxylase. GABA accumulates in rice at low temperatures and is a typical emergency metabolite. In addition to its osmoregulatory properties, GABA, which is widely found in plants, boosts antioxidant enzymatic activity and effectively scavenges free radicals [31]. The results of our study indicated that low-temperatures treatments (LT and LT+N2) were conducive to the conversion of L-glutamine to GABA and elevated the antioxidant enzyme activities in rice. The succinic semialdehyde content increased during storage at LT treatment. There was a significant decrease in the succinic aldehyde semialdehyde levels; succinic semialdehyde was converted mainly to succinate. High levels of succinic semialdehyde, which is harmful to plant tissues, increase the levels of reactive oxygen intermediates and reduce plant growth [32]. The application of low-temperature treatments (LT and LT+N_2_) resulted in a reduction in the damage to rice caused by the accumulation of succinic semialdehyde.

Its catabolic pathway plays a central role in plant immunity and is activated when plants are exposed to microbial pathogens [33]. L-lysine catabolism produces the immunological signal pipecolic acid, which enhances plant defense responses. It is a critical regulator of systemic acquired tolerance in plants. L-lysine decreased in response to salt stress, making it a promising candidate to support callus formation and potentially increase osmotic pressure to facilitate water uptake from neighboring cells. The activity of reactive oxygen species-related enzymes is regulated by L-lysine [33]. Therefore, it can be hypothesized that L-lysine has an antioxidant-inducing effect on rice and reduces the damage to the function of the rice tissues. The MDA level was a measure of the degree of oxidative damage to the cell membrane. L-lysine regulation decreased the antioxidant enzyme activity and the active scavenging capacity of rice when it was stressed. This may explain the increase in MDA content under routine storage conditions. In plants, L-serine also plays an important role in the stress response to environmental stressors such as high salt levels, flooding, and low temperatures [34]. The level of L-serine in LT+N_2_-treated rice was significantly up-regulated compared to CS. It suggested that L-serine improved the stress tolerance under LT conditions.

The metabolic differences in rice under different storage conditions were evident. Rice accumulated more amygdalin, L-asparagine, glutamate, and GABA under low-temperature conditions. Under a nitrogen-modified atmosphere, L-serine was stored, whilst L-lysine was reduced to protect itself from environmental stress.

#### 3.3.2. Carbohydrate Metabolism

Lactose can be employed as a low-concentration signaling molecule in plants, participating in the regulation of plant growth and development, as well as adversity adaptation. Furthermore, lactose constitutes a significant component of the plant cell wall, participating in the synthesis and degradation processes of the plant cell wall. G-1-P plays a pivotal role in the processes of energy storage and conversion. In plant cells, G-1-P can be converted to starch in a series of enzymatic steps involving multiple intermediates. Low temperatures impede the breakdown of lactose and stimulate the catabolism of G-1-P. The application of LT and LT+N_2_ treatments increased the enzyme activity, thereby facilitating the conversion of G-1-P to starch during the reaction. Furthermore, rice under low-temperature treatments (LT and LT+N_2_) exhibited an increased lactose content, collectively resulting in the sustained maintenance of the rice grain cell wall morphology.

The discovery of raffinose as an antioxidant is recent. Raffinose is a member of a group of soluble carbohydrates known as RFOs, which have been researched extensively for their ability to improve abiotic stress tolerance. Raffinose can protect plants from freezing and cold damage [35]. Raffinose is not only an osmotic protection and cell membrane stabilizer but it is also a scavenger of reactive oxygen species. It plays a novel role in protecting cellular metabolism, particularly from oxidative damage caused by salt, cold, or drought [36]. Melibiose is a disaccharide that is in vivo indigestible by humans. The compound can be employed as a valuable additive in functional foods and pharmaceuticals intended for humans, with the objective of maintaining and enhancing their well-being. In our experimental results, the melibiose and raffinose contents in the low-temperature treatment (LT and LT+N_2_) groups were elevated, suggesting that the use of low-temperature treatment promoted the synthesis of raffinose and melibiose. Thus, the low-temperature treatment was more efficacious in safeguarding the integrity of rice cell membranes, effectively preventing oxidative damage, and maintaining the nutritional value of rice grains.

It has been shown that the level of D-gluconate has a negative correlation with the germination and seed vigor of rice grains. D-gluconate was high in rice grains with low vigor. D-gluconate may be a candidate marker for seed senescence [37]. Low levels of D-gluconate may indicate increased resistance in rice. D-ribose and 2-deoxy-D-ribose 5-phosphate were derived from D-gluconate. D-ribose is a naturally occurring monosaccharide that is a structural component of DNA and RNA and is a component of ATP. D-ribose causes cellular damage, and it induces AGE accumulation and ROS production. ROS accumulation leads to oxidative DNA damage and, thus, mitochondrial apoptosis [38]. The low contents of D-ribose and 2-deoxy-D-ribose 5-phosphate in rice under low-temperature treatments (LT and LT+N_2_) indicated the elevated antioxidant capacity of rice.

#### 3.3.3. Flavone and Flavonol Biosynthesis

Most of the health benefits of flavonoids are attributed to their antioxidant and chelating abilities. Kaempferol demonstrates a range of physiological activities, including antioxidative effects. The study showed that D-ribose caused the accumulation of advanced glycation end products (AGEs) and reactive oxygen species (ROS) in mesangial cells, resulting in mitochondrial apoptosis. However, kaempferol was found to mitigate these effects, and its protective effect may be linked to the restoration of autophagy [38]. In our results, the N_2_ treatment induced elevated levels of kaempferol, which could slow down mitochondrial apoptosis by reducing the accumulation of AGEs and ROS, thereby protecting rice from damage.

In our study, the CS and N_2_ groups showed an increase in kaempferol content at high temperatures. This increase may be attributed to cellular damage caused by repairing D-ribose. Quercitrin exhibits an inhibitory activity against aldose reductase [39]. Quercitrin is a widely distributed antioxidant biomolecule found in edible plants. It has anti-cancer properties and acts as an antioxidant [40]. Quercetin has anti-melanogenic efficacy. In our study, quercitrin was up-regulated in rice following the LT+N_2_ treatment, indicating that rice grains in the LT+N_2_ treatment group had more potent antioxidant properties and could better withstand environmental stress.

Natural flavonoid compounds, including luteolin, exhibit free radical scavenging and antioxidant activities by reacting with phenolic hydroxyl groups to form stable semiquinone radicals that terminate the free radical chain reaction [41]. Furthermore, luteolin compounds have the potential to increase the enzymatic activities of antioxidant enzymes such as SOD and CAT, and luteolin has a superior antioxidant activity compared to quercetin [42]. Luteolin has been found to possess anti-radiation properties and can help reduce radiation-induced ROS damage in plants and animals [43]. Luteolin is a compound that helps defend the rice grain against high-temperature stress. Our results suggest that the production of luteolin is activated under N_2_ treatment conditions, enhancing the defense function in rice.

## 4. Conclusions

In this study, the effects of LT and N_2_ treatments on the storage quality and processing characteristics, as well as the metabolites of rice during storage, were investigated. In addition, non-targeted metabolomics was used to screen for differential metabolites and determine the optimal storage conditions in rice grain. The contents of FAV and MDA as well as the electrical conductivity and CAT activity increased at a slower rate in rice stored under LT+N_2_ conditions toward the end of the storage period. Furthermore, the LT+N_2_ treatment was more effective at inhibiting the deterioration of the color of the paddy and preserving the pasting properties of the rice. By regulating the metabolisms of amino acids, carbohydrates, and flavonoids, the LT+N_2_ treatment increased the oxidative stress capacity of rice, helping to maintain the vigor of the grain and retarding its spoilage. Thus, it is concluded that LT+N_2_ treatment is an effective method for delaying quality deterioration in rice storage.

## Figures and Tables

**Figure 1 foods-13-02968-f001:**
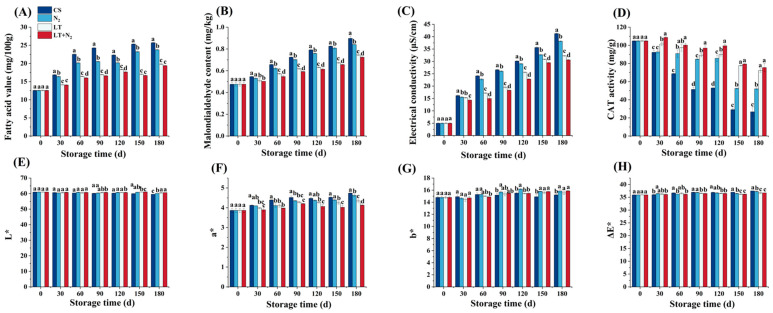
Changes in the storage characteristics of rice samples that were stored under various conditions (CS, LT, N2, and LT+N_2_). (**A**) Changes in FAV of rice samples. (**B**) Changes in MDA contents of rice samples. (**C**) Changes in electric conductivity of rice samples. (**D**) Changes in CAT activity of rice samples. (**E**) Changes in L* of rice samples. (**F**) Changes in a* of rice samples. (**G**) Changes in b* of rice samples. (**H**) Changes in ∆E* of rice samples. In the graph, different letters indicated significant differences (*p* < 0.05).

**Figure 2 foods-13-02968-f002:**
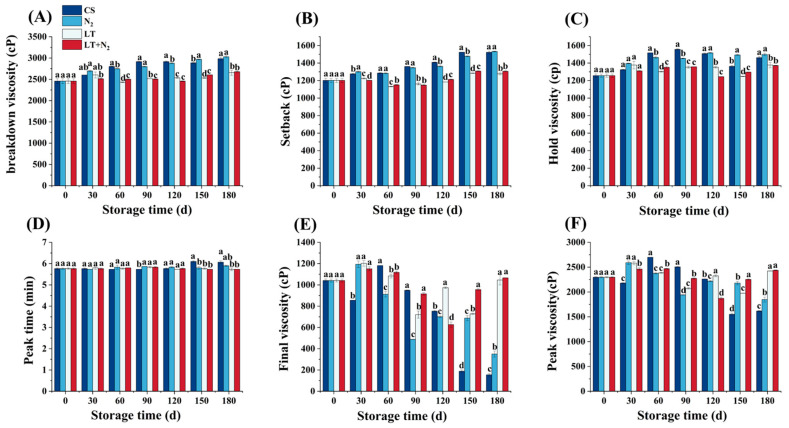
Changes in the processing characteristics of rice samples that were stored under various conditions (CS, LT, N2, and LT+N_2_). (**A**) Changes in breakdown viscosity of rice samples. (**B**) Changes in setback of rice samples. (**C**) Changes in hold viscosity of rice samples. (**D**) Changes in peak time of rice samples. (**E**) Changes in final viscosity of rice samples. (**F**) Changes in peak viscosity of rice samples. In the graph, different letters indicated significant differences (*p* < 0.05).

**Figure 3 foods-13-02968-f003:**
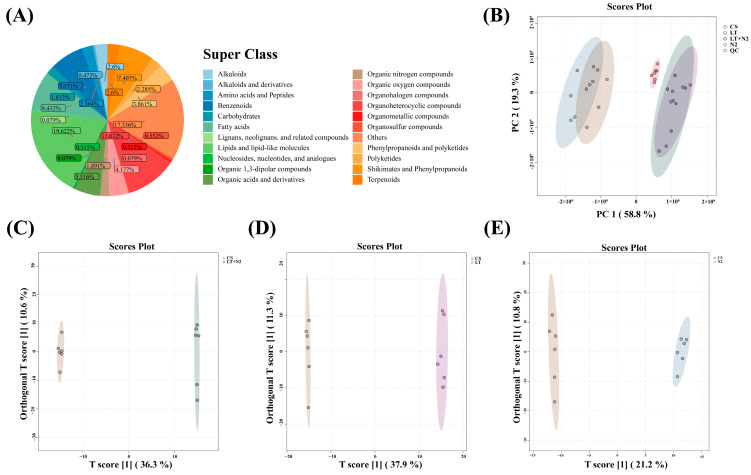
(**A**) Pie plot and score plots of rice sample metabolite classes and compositions. PC 1 and PC 2 represented the scores of the first- and second-ranked principal components, respectively. (**B**) PCA score plots of CS, LT, N_2_, and LT+N_2_. (**C**) OPLS-DA score plots of CS and LT+N_2_. (**D**) OPLS-DA score plots of CS and LT. (**E**) OPLS-DA score plots of CS and N_2_. The horizontal coordinate t (1) P represents the predicted first principal component score, showing sample component differences. The vertical coordinate t (1) O represents the orthogonal principal component score. The OPLS-DA model was used to screen for variable weights (VIP values) > 1.

**Figure 4 foods-13-02968-f004:**
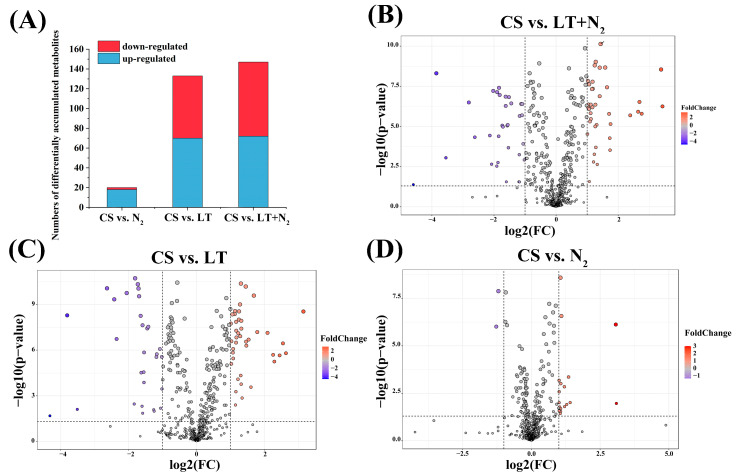
(**A**) Numbers of differentially accumulated metabolites. (**B**) The volcano plot of CS vs. LT+N_2_. (**C**) The volcano plot of CS vs. LT. (**D**) The volcano plot of CS vs. N_2_. Upregulated metabolites are marked in red, and down-regulated metabolites are marked in blue. The screening conditions were VIP ≥ 1, fold change (FC) ≥ 2, and *p*-value < 0.05.

**Figure 5 foods-13-02968-f005:**
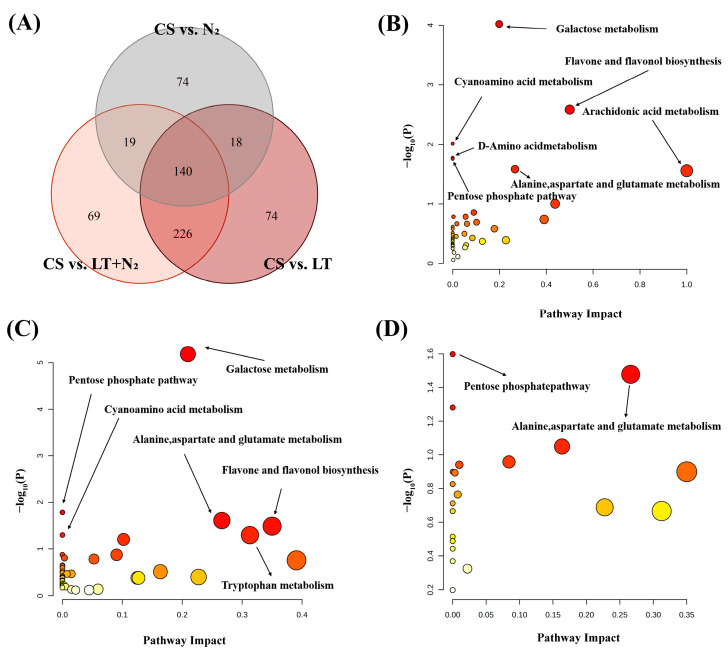
(**A**) Venn diagram indicating overlapping metabolites in different groups. Pathway impact analysis displaying changing metabolisms in (**B**) CS vs. N_2_; (**C**) CS vs. LT; and (**D**) CS vs. N_2_.

**Figure 6 foods-13-02968-f006:**
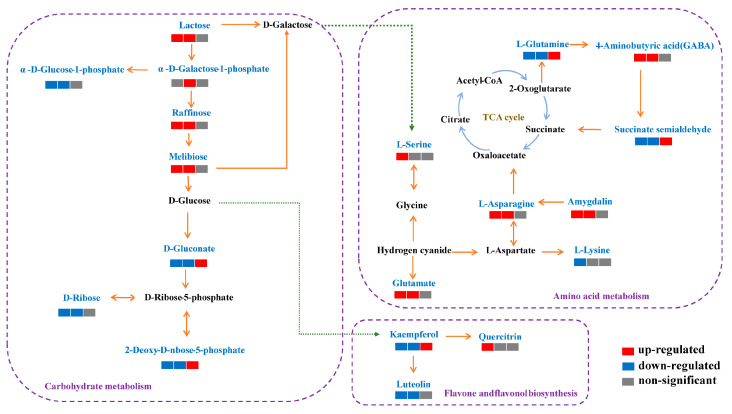
Relation of metabolites in rice under different conditions. Colors indicate up- or down-regulated metabolites for each treatment. The figure shows a comparison of the different storage conditions, from left to right as follows: CS vs. LT+N_2_, CS vs. LT, and CS vs. N_2_. Metabolites that are up-regulated in rice are highlighted in red, while those that are down-regulated are highlighted in blue. The color gray indicates no significant change in metabolites.

## Data Availability

The original contributions presented in the study are included within the article/Appendix A; further inquiries can be directed to the corresponding author.

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
