# Peer review of "Changes in Quality Characteristics and Metabolite Composition of Low-Temperature and Nitrogen-Modified Atmosphere in Indica Rice during Storage"

_foods, 2024, doi:10.3390/foods13182968_

Round 1

Reviewer 1 Report

Comments and Suggestions for Authors

To verify the plant species used and the consistency of their quality, it is crucial to address the identification of the rice cultivar (variety) used in this study. Please elaborate on the methods or protocols employed to ensure that the plants acquired for the study accurately corresponded to the specified species.

It is of importance to measure antioxidant property of the rice during storage via performing assays, such as FRAP, DDPH, and ABTS.

How about the total phenolic content (TPC) of the samples during storage under different treatments?

Among the differentially accumulated metabolites, which group contributed the most to the quality maintenance of the samples during storage? 

How could the low temperature and nitrogen treatment delay the quality deterioration in the samples during storage? a schematic overview as a figure would provide readers with a better understanding of the mechanism(s) involved. In addition, the possibility of the mechanism(s) involved at the transcriptome level should also be included and discussed. 

Comments on the Quality of English Language

Minor revision is required. 

Author Response

Dear reviewer:

We are grateful for your comments on the Manuscript foods-3186516 and the opportunity you give us to revise this manuscript.

We have revised the manuscript carefully. The changes can be seen with red color in the revised manuscript. The specific changes were listed below.

  1. To verify the plant species used and the consistency of their quality, it is crucial to address the identification of the rice cultivar (variety) used in this study. Please elaborate on the methods or protocols employed to ensure that the plants acquired for the study accurately corresponded to the specified species.

Response: Thank you for your suggestion. The part was rewritten in Lines 97-98.

  1. It is of importance to measure antioxidant property of the rice during storage via performing assays, such as FRAP, DDPH, and ABTS.

Response: Thank you for your suggestion. In our experimental studies, the measurement of CAT enzyme activity can be used to indicate antioxidant capacity. Additionally, the level of MDA, a product of lipid peroxidation of cell membranes, can be used as an indicator of the degree of oxidative stress [1]. Thank you for giving us good ideas, in the next step we will scrutinize the antioxidant mechanism during rice storage.

  1. How about the total phenolic content (TPC) of the samples during storage under different treatments?

Response: Thank you for your suggestion. The total phenol content was not directly determined in the experiment. But there was a relevant representation in metabolomics. Rice grains are known to contain a variety of phenolic compounds, including flavonoids, phenolic acids, tannins, and others, which exist in both bound and free forms. The results of the metabolism section of our paper indicate that six phenolic acids were metabolized differentially by CS vs LT+N2, and eight phenolic acids were metabolized differentially by CS vs LT. Only dimethyl phthalate was down-regulated, while the remainder of the substances were up-regulated in the comparison between the two groups. A total of eight phenolic acids were identified, with Salicylic acid, Methyl vanillate, Methyl isovanillate exhibiting down-regulation. The majority of flavonoids exhibited a notable decline in expression, while the LT+N2 combination treatment demonstrated the capacity to stimulate the upregulation of epigallocatechin, which exhibited downregulation when acting alone. Conversely, the LT+N2 combination treatment was observed to promote the up-regulation of quercitrin.

  1. Among the differentially accumulated metabolites, which group contributed the most to the quality maintenance of the samples during storage?

Response: Thank you for your suggestion. The part was rewritten in Lines 339-347.

  1. How could the low temperature and nitrogen treatment delay the quality deterioration in the samples during storage? a schematic overview as a figure would provide readers with a better understanding of the mechanism(s) involved.

Response: Thank you for your suggestion. In Figure 6 of the article, we represent the mechanism of key pathway metabolites during rice storage. The figure shows a comparison of the different storage conditions, from left to right: CS vs LT+N2, CS vs LT and CS vs N2. Metabolites that are up-regulated in rice are highlighted in red, while those that are down-regulated are highlighted in blue. The color gray indicates no significant change in metabolites.

  1. In addition, the possibility of the mechanism(s) involved at the transcriptome level should also be included and discussed. 

Response: Thank you for your suggestion. We agree that more studies would be useful to understand the details of interaction and enhancement. At this point, we do not have the necessary tool set to study the transcriptome level. In the future, we hope to employ techniques to determine transcriptome level.

Best Wishes

Yours sincerely

Yan Zhao

References:

[1] Zeng, F., Qiu, B., Wu, X., Niu, S., Wu, F., & Zhang, G. (2012). Glutathione-Mediated Alleviation of Chromium Toxicity in Rice Plants. Biological Trace Element Research, 148(2), 255-263. https://doi.org/10.1007/s12011-012-9362-4.

Reviewer 2 Report

Comments and Suggestions for Authors

Congrats for your special and high quality work. 

Author Response

Dear reviewer:

Thank you for meticulously reviewing my Manuscript foods-3186516 and giving it such a high rating. I am deeply honored and encouraged to learn that you believe the thesis needs no revision.

Your recognition means a lot to me, not only as an affirmation of my past efforts but also as an incentive for my future research work. I will continue to dedicate myself to in-depth research and academic exploration and strive to make more contributions to the academic community.

Thank you again for your valuable time and professional input. I look forward to the opportunity to work with you again in the future and to work together to advance academic research.

Best Wishes

Yours sincerely

Yan Zhao

Reviewer 3 Report

Comments and Suggestions for Authors

It is interesting to combine the treatment of LT and N2 during storage. The treatments (CS, LT, N2, and LT+ N2 ) used were well proposed.

The title of the article: should be improved.

Abstract: Low-temperature (LT). Please, indicate the specific temperature and N2% used. It is also necessary to specify how long the product was stored and the relative storage humidity.180 days?

LT+N2 treatment can be applied on an industrial scale. What is its feasibility?

Materials and Methods:

Rice storage - change 6 months by 180 days. 20 ± 1 ℃ is an LT to store rice? Reference. How did you define these storage conditions?

Malondialdehyde content (MDA) cite correctly  Li et al. (2022) [20]

What is the electrical conductivity at 20 ℃ ?

Pasting properties of rice -What was the grain size (granulometry) of the rice flour used?

Figure 5 could be supplementary material and should improve the quality.

The conclusion seems fine 

Microbiological quality must be reported with at least data from the beginning and end of storage.

Comments on the Quality of English Language

 Minor editing of English language required.

Author Response

Dear reviewer:

We are grateful for your comments on the Manuscript foods-3186516 and the opportunity you give us to revise this manuscript.

We have revised the manuscript carefully. The changes can be seen with red color in the revised manuscript. The specific changes were listed below.

  1. Improvement of article titles

Response: Thank you for your suggestion. The title was rewritten in Lines 2-4.

  1. Line 16: Low-temperature (LT). Please, indicate the specific temperature and N2% used. It is also necessary to specify how long the product was stored and the relative storage humidity.180 days?

Response: Thank you for your suggestion. The sentence was rewritten in Lines 14-16.

  1. Line 113: change 6 months by 180 days.

Response: Thank you for your suggestion. The sentence was rewritten in Line 110.

  1. 20 ± 1 â„ƒ is an LT to store rice? Reference. How did you define these storage conditions?

Response: In China's large-scale grain storage facilities, the average grain temperature is maintained at 15 ℃ or below throughout the year. Moreover, the temperature at the maximum localized point of the grain within the barn should not exceed 20 degrees Celsius. In a practical grain silo context, the financial outlay rises markedly with each degree of temperature reduction within the silo. Therefore, 20 ℃ was selected as the experimental setup temperature.

  1. Malondialdehyde content (MDA) cite correctly Li et al. (2022) [20]

Response: Thank you for your suggestion. The sentence was rewritten in Line 125.

  1. What is the electrical conductivity at 20 ℃?

Response: Thank you for your suggestion. The part was rewritten in Figure 1C and Line 138.

  1. 7. Line 156: Pasting properties of rice -What was the grain size (granulometry) of the rice flour used?

Response: Thank you for your suggestion. The sentence was rewritten in Line 153.

  1. 8. Figure 5 could be supplementary material and should improve the quality.

Response: Thank you for your suggestion. We have submitted a higher version of Figure 5 in a supplementary document. The sentence was rewritten in Lines 505-506.

  1. Microbiological quality must be reported with at least data from the beginning and end of storage.

Response: Thank you for your suggestion. In our experiments, we observed that nitrogen and low temperature treatments affected the microorganisms. We will describe the changes in microorganisms in the next article.

  1. Minor editing of English language required.

Response: Thank you for your suggestion. We tried our best to improve the manuscript and made some changes to the manuscript. These changes will not influence the content and framework of the paper. Here, we did not list the changes but marked them in red in the revised paper; we appreciate the Reviewers’ warm work earnestly and hope that the correction will meet with approval.

Round 2

Reviewer 1 Report

Comments and Suggestions for Authors

Revision is fine.